# The Promise and Challenges of Cyclic Dinucleotides as Molecular Adjuvants for Vaccine Development

**DOI:** 10.3390/vaccines9080917

**Published:** 2021-08-17

**Authors:** Hongbin Yan, Wangxue Chen

**Affiliations:** 1Department of Chemistry, Brock University, St. Catharines, ON L2S 3A1, Canada; 2Human Health and Therapeutics Research Centre, National Research Council Canada, Ottawa, ON K1A 0R6, Canada; 3Department of Biological Sciences, Brock University, St. Catharines, ON L2S 3A1, Canada

**Keywords:** cyclic dinucleotide, c-di-GMP, bacterial second messenger, immunostimulation, vaccine adjuvant

## Abstract

Cyclic dinucleotides (CDNs), originally discovered as bacterial second messengers, play critical roles in bacterial signal transduction, cellular processes, biofilm formation, and virulence. The finding that CDNs can trigger the innate immune response in eukaryotic cells through the stimulator of interferon genes (STING) signalling pathway has prompted the extensive research and development of CDNs as potential immunostimulators and novel molecular adjuvants for induction of systemic and mucosal innate and adaptive immune responses. In this review, we summarize the chemical structure, biosynthesis regulation, and the role of CDNs in enhancing the crosstalk between host innate and adaptive immune responses. We also discuss the strategies to improve the efficient delivery of CDNs and the recent advance and future challenges in the development of CDNs as potential adjuvants in prophylactic vaccines against infectious diseases and in therapeutic vaccines against cancers.

## 1. Introduction

Host innate immunity is the first line of defence against invading microbial pathogens and cancers and also plays a critical role in the development of autoimmune diseases. Innate immune cells, such as macrophages, granulocytes, dendritic cells (DCs), natural killer (NK) cells, and mast cells, contain specific pathogen-recognition receptors (PRRs), which sense pathogen-associated molecular patterns (PAMPs) and induce the production of proinflammatory cytokines/chemokines and subsequently activate the adaptive immune response. A variety of PRRs has now been identified and characterized, including the Toll-like receptors (TLRs), nucleotide-binding oligomerization domain (NOD)-like receptors (NLRs), and Retinoic acid-inducible gene (RIG)-I-like receptors (RLRs). In addition, cyclic dinucleotides (CDNs), including bacterial second messengers bis-(3′,5′)-cyclic diguanosine monophosphate (c-di-GMP) and bis-(3′,5′)-cyclic diadenosine monophosphate (c-di-AMP), 3′5′-3′5′ cyclic GMP-AMP (3′3′-cGAMP) produced by *Vibrio cholerae*, and metazoan second messenger 2′5′-3′5′ cyclic GMP-AMP (2′3′-cGAMP, hereafter cGAMP), bind to an endoplasmic reticulum (ER)-associated sensor termed as a stimulator of interferon genes (STING) [1,2,3,4]. The finding that CDNs can trigger the innate immune response in eukaryotic cells through the STING signalling pathway, leading to the type I interferons (IFN) production, has prompted the extensive research and development of CDNs as potential vaccine adjuvants and agents for cancer immunotherapy [5].

In this review, we summarize the structure and chemical synthesis, the biological characteristics, the current status of CDN vaccine adjuvant research, including their superior adjuvant activities, in vivo mode of action, and application in the development of infectious disease and cancer vaccines. We also examine the strategies to improve the efficient delivery of CDNs and future challenges in advancing the lead CDN adjuvant candidate into clinical trials and ultimately in human clinical use.

## 2. Discovery and Chemical Structure of CDNs

In their investigation of the regulation of cellulose synthesis in *Acetobacter xylinum* (now classified as *Gluconacetobacter xylinus*), Benziman and co-workers discovered a cyclic oligonucleotide that appeared to be an activator of cellulose synthase [6,7]. This activator was subsequently identified in the same laboratory as bis-(3′→5′)-cyclic diguanylic acid (also known in the literature as 3′,5′-cyclic diguanylic acid, c-di-GMP, or c-diGMP, Figure 1), with the mass spectrometric and NMR data in agreement with samples generated by enzymatic synthesis from guanosine triphosphate (GTP) catalyzed by diguanylate cyclase [8]. The crystal structure of chemically synthesized c-di-GMP was subsequently resolved by Rich and co-workers [9].

The adenylyl analog of c-di-GMP, c-di-AMP, together with the diadenylate cyclase activity of the nucleotide-binding domains of the DNA-integrity scanning protein A (DisA), were discovered in 2008 [10]. The DisA protein was shown to control a sporulation checkpoint in *Bacillus subtilis* in response to DNA double-strand breaks. Since this discovery, the roles of c-di-AMP in bacterial physiology were rapidly expanded [11,12].

A guanosine/adenosine hybrid CDN, 3′3′-cGAMP (3′5′-3′5′ cyclic GMP-AMP or c(GpAp)), was also found in a *V. cholerae* strain expressing dinucleotide cyclase DncV [13], which is involved in intestinal colonization and chemotaxis of *V. cholerae*.

In two back-to-back publications in *Science* in 2013, Chen and co-workers reported the roles of cyclic GMP-AMP synthase in the activation of the type I interferon pathway [14], and the identification of 2′2′-cyclic GMP-AMP (2′2′-cGAMP) [15] as the endogenous ligand for cyclic GMP-AMP synthase. Very shortly after the publication of these original results, it was determined that the endogenous ligand is, in fact, 2′3′-cyclic GMP-AMP (2′3′-cGAMP) [16].

In the ensuing few years since its discovery, c-di-GMP attracted very little attention until around 2004/2005 when its roles as a bacterial second messenger were recognized, and the field quickly expanded with the potential of this cyclic dinucleotide and its analogs as immunostimulatory agents and vaccine adjuvants. The rapidly growing interest in cyclic dinucleotides is clearly reflected by the expansion of the number of published records in the last two decades (Figure 2).

## 3. c-di-GMP as a Universal Bacterial Second Messenger

Intracellular levels of c-di-GMP are regulated through the actions of diguanylate cyclases (DGC) and phosphodiesterases (PDE), usually as a response to an environmental stimulus, i.e., first messenger. In signalling cascades involving c-di-GMP, binding of this second messenger with effector molecules, which are usually proteins and, in some cases, RNA such as riboswitches, allows for the regulation of the processes mediated by these effector molecules. Most DGCs contain a conserved GGDEF domain, although DGC cyclase enzymes with GGEEF, AGDEF, and GGDEM domains have also been identified [17]. PDEs, on the other hand, typically contain EAL (or less frequently HY-GYP) domains. Interestingly, numerous other proteins containing these domains are enzymatically inactive.

In the last 15 years or so, significant progress has been made toward the identification of effector molecules that are key to the biological processes or phenotypes regulated by c-di-GMP. The first such effector protein identified was the cellulose synthase from *G. xylinus* (formerly *A. xylinum*) [8,18], where the cellulose synthesis is activated by c-di-GMP. Almost 20 years after the discovery of the regulation of cellulose synthase activity via the allosteric binding with c-di-GMP, the PilZ domain was identified to be responsible for the binding to c-di-GMP [19]. Indeed, PilZ was subsequently found to be a common c-di-GMP binding domain among bacterial species, regulating bacterial motility [20,21,22,23] and virulence [24]. Other proteins such as the PelD protein from *Pseudomonas aeruginosa* [25], LapD protein from *P. fluorescens* [26], the transcription factor VpsT from *V. cholera*, the transcription factor FleQ (flagellar transcriptional regulator) from *P. aeruginosa* [27], and the Bcam1349 protein from *Burkholderia cenocepacia* [28] have also been identified as effector proteins in c-di-GMP-mediated processes. Furthermore, riboswitches [29,30] and self-splicing ribozyme [31] have been shown to be another class of effector molecules. Located in the 5′-untranslated upstream of the open reading frames of DGC and PDE proteins, binding of these riboswitches to c-di-GMP can serve as a regulator for the expression of DGC and PDE proteins.

Taken together, c-di-GMP mediated signalling networks have been shown to affect the regulation of a wide range of biological processes and phenotypes in bacteria (Figure 3), enabled by the binding of this second messenger to an effector molecule.

## 4. c-di-GMP and Analogs Generated to Date

While c-di-GMP can be isolated from bacterial cultures [32], this approach is very limited in the amount of accessible material. Furthermore, the generation of modified c-di-GMP via this approach is usually unfeasible. With this in mind, a number of methods have been described toward the synthesis of c-di-GMP and analogs, allowing for access to a relatively large amount of material for structural, bacterial, and immunological applications.

The vast majority of CDN analogs reported to date are based on phosphate or modified phosphate backbone in nature. To date, CDNs with all possible combinations of canonical nucleobases have been synthesized [33]. Appendix A summarizes the c-di-GMP analogs reported to date that feature modifications in nucleobases, various sugar residues (e.g., ribose, deoxyribose, and 2′-modified ribose), phosphate, and phosphorothioate. A selection of CDNs containing both 2′-5′ and 3′-5′ internucleoside phosphate/phosphorothioate linkages (Appendix A) have also been made available through chemical synthesis. Among the chemistry reported for the synthesis of c-di-GMP analogs, some features are of important consideration: versatility of the chemistry, ease to scale up the process, and to purify the final product. It is worth noting that c-di-GMP [34,35], c-di-araGMP and c-di-dGMP [36], and other analogs [37], 3′3′-cGAMP and 2′3′-cGAMP [38,39] and their analogs [39,40] have also been generated enzymatically.

Furthermore, CDN analogs where the phosphate backbone is replaced with bis-carbamate, amide, urea, thiourea, carbodiimide, and triazolyl have also been synthesized (Appendix A) to explore their applications in bacterial physiology. In addition, conjugates of c-di-GMP with other molecular entities (Appendix A) have also been synthesized to study biological processes involving this bacterial second messenger. These conjugates are particularly useful to visualize c-di-GMP when a fluorophore is covalently attached or to pull-down molecules that bind to c-di-GMP when biotin is conjugated to c-di-GMP.

## 5. Immunostimulatory Function of CDNs in Host Innate Immune Responses

It is now well established that CDNs activate innate immunity by directly binding to STING to induce the production of type I IFN and inflammatory cytokines via TANK-binding kinase 1 (TBK1) and interferon regulatory factor 3 (IRF3) pathway (Figure 4) [41,42]. However, alternative pathways also exist. McFarland et al. have identified the oxidoreductase RECON as a cytosolic sensor for bacterial CDNs in modulating the inflammatory gene activation via its antagonistic effects on STING and nuclear factor kappa B (NF-κB) [43]. In addition, it was reported that c-di-GMP could induce antigen-specific antibody and balanced T helper type 1 (Th1)/Th2/Th17 cell responses through a novel IFN-I-independent, STING-NF-κB-tumour necrosis factor (TNF)-α pathway, which is distinct from STING-mediated DNA adjuvant activity that requires IFN-I, but not TNF-α, production [44]. The critical role of TNF-α signalling in this pathway was confirmed in studies using TNFR1(−/−) mice. Moreover, Chang et al. have recently suggested A (2B) adenosine receptors in the intestinal epithelium as a potential pathway for extracellular CDNs to enter intracellular cytosols [45].

### 5.1. In Vitro Immunostimulatory Functions of CDNs

The in vitro immunostimulatory function of CDNs has been studied in cultured human and murine cells such as immature DCs and macrophages. These studies have identified DCs and macrophages as the major target cells of c-di-GMP and c-di-AMP. Analyses of DC subsets revealed conventional DCs as principal responders to c-di-AMP stimulation [47]. Both c-di-GMP and c-di-AMP were able to promote the activation and maturation of murine and human DCs [48,49] and induce the production of IFN-β and interleukin (IL)-6, but not TNF, in RAW264.7 macrophage cells and mouse primary bone marrow-derived macrophages (BMDMs) [50]. In cultured human immature DCs, c-di-GMP upgraded the surface expression of costimulatory molecules CD80 and CD86, maturation marker CD83, and MHC class II. c-di-GMP treatment also altered the expression of chemokine receptors C-C Motif Chemokine Receptor 1 (CCR1), CCR7, and C-X-C Motif Chemokine Receptor 4 (CXCR4), and increased the production of IL-12, IFN-γ, IL-8, CC chemokine ligand 2 (CCL2), C-X-C motif chemokine ligand 10 (CXCL10), and CCL5 by DCs. Studies on M1- and M2-polarized THP-1 and murine BMDMs showed that both murine and human M1-polarized macrophages showed increased STING expression and treatment with a STING agonist (DMXAA) reprogramed M2 macrophages toward an M1-like subtype [51].

c-di-GMP-matured DCs showed enhanced T-cell stimulatory activity [48,49]. Stimulation of bone marrow-derived DCs (BMDCs) with CDNs induced marked metabolic products (nitric oxide and the cytokine BAFF) that are different from those induced by TLR ligands [52]. The BAFF production appears to be correlated with the improved (i.e., more rapid and persistent) antibody responses in both aged and young mice [52]. Recent studies have also shown that c-di-GMP differentiated lung monocyte-derived DCs (MoDCs) into Bcl6^+^ mature MoDCs and promoted lung memory T helper cells through a process mediated by soluble TNF. MoDCs are essential for c-di-GMP-induced lung mucosal responses in this model and are responsible for lung IgA responses, but they are dispensable for vaccine-induced systemic immune responses [53]. In addition to type 1 IFN, physiologically relevant levels of c-di-AMP and c-di-GMP also stimulated a robust IL-1β production in murine BMDMs by the activation of NLRP3 inflammasome through a unique pathway associated with the recognition of intracellular infections, but independent of the STING pathway [54].

CDNs (such as c-di-GMP) also synergized with B cell receptor signals to directly activate B cells and promote antibody responses in a STING-dependent manner [55]. Similarly, costimulation of T cells with cGAMP and TCR ligands induced IFN-I production and regulated the growth and function of T cells through the mTORC1 pathway [56]. Studies in unfractionated human peripheral blood mononuclear cell (PBMC) cultures have shown that both 2′3′-cGAMP and 3′3′-cGAMP were highly potent in driving the expansion and maturation of functional antitumor or antiviral antigen-specific CD8^+^ T cells via the induction of type I IFNs [57,58,59]. 3′3′-cGAMP-primed HIV-1-specific CD8^+^ T cells produced higher levels of perforin and granzyme B expressions than lipopolysaccharides (LPS)- or TLR7/8 agonist R848-primed ones and were more potent to suppress HIV-1 [58,59]. c-di-AMP and, to a lesser extent, c-di-GMP also played a divergent role in regulating *Porphyromonas gingivalis* LPS-induced cytokine responses in human gingival keratinocytes [60]. c-di-AMP treatment increased the extracellular MCP-1 and vascular endothelial growth factor (VEGF) levels and intracellular IL-1 receptor antagonist level and restored decreased extracellular IL-8 levels induced by LPS. In the presence of LPS, c-di-GMP increased extracellular VEGF level, whereas c-di-AMP suppressed intracellular IL-1β levels [60].

### 5.2. In Vivo Immunostimulatory Functions of CDNs

In corroboration with the in vitro results, experimental studies in animal models have confirmed the immunostimulatory function of CDNs in inducing host innate immune responses and enhancing the adaptive immune responses. Systemic (intraperitoneal, i.p.) or mucosal (intranasal, i.n.) administration of mice with c-di-GMP induced an acute and transient local recruitment of inflammatory cells (neutrophils and macrophages) and activation of DCs [48,61,62,63]. Intranasal c-di-GMP administration also induced the production of chemokines CXCL1, CCL2, CCL3, CXCL2, and CCL5 [63]. Similarly, systemic or local administration of cGAMP induced high levels of type I IFNs in mice but minimal inflammatory gene expression [64]. Intranasal administration of c-di-GMP in mice greatly enhanced the antigen uptake by CD11c^+^ cells through both pinocytosis and receptor-mediated endocytosis, which was dependent on the increased STING expression [65]. In this regard, c-di-GMP selectively activated DCs with high pinocytosis efficiency and induced the production of IL-12p70, IFNγ, IL-5, IL-13, IL-23, IFNλ, and IL-6, but interestingly not IFNβ production, in this model [65].

The immunostimulatory properties of CDNs have been explored for their potential as novel immunotherapeutics for the treatment of infections, cancer, and other immunological disorders. Intranasal, subcutaneous (s.c.), intramuscular (i.m.), i.p., and intramammary administration with CDNs induced robust innate immune responses and enhanced host resistance against infection with a wide range of pathogenic bacteria, such as *Bordetella pertussis*, *Acinetobacter baumannii*, *Staphylococcus aureus*, *Klebsiella pneumoniae*, *Clostridium perfringens*, and various serotypes of *Streptococcus pneumoniae* [48,62,63,66,67,68]. Depletion of neutrophils abolished the protective role of c-di-GMP in the mouse model of *Acinetobacter* pneumonia, underscoring the importance of neutrophils as the critical effector cell in c-di-GMP-induced protection [63]. In addition, studies in a chicken model of *C. perfringens* infection have shown the promise of CDNs as an alternative to antibiotics without increasing the selection pressure for some β-lactamase genes or altering the commensal bacterial population [67].

Recent studies have shown that systemic treatment of mice with cGAMP inhibited the replication of genital herpes simplex virus (HSV) 2 and improved the clinical outcome of infection [64]. More importantly, direct application of CDNs, but not TLR agonists, on the genital epithelial surface increased local IFN activity and conferred total protection against disease even in immunocompromised mice without causing overt systemic responses [64]. Moreover, therapeutic administration of encapsulated-cGAMP microparticles (MPs), but not soluble cGAMP, protected mice from the development and relapse of experimental allergic encephalomyelitis in an IFN-I-dependent and -independent mechanism, and the therapeutic outcome was more effective than recombinant IFN-β [69]. In this regard, in vitro studies of PBMCs from relapsing-remitting multiple sclerosis patients showed that cGAMP MPs promoted both IFN-I and the immunoregulatory cytokines IL-27 and IL-10 responses [69].

The identification of the critical contribution of STING to antitumor immunity has generated a great enthusiasm to explore the intratumoral administration of STING agonists as standalone cancer immunotherapy or in combination with other cancer immunotherapeutic modalities such as immune checkpoint inhibitors and chimeric antigen receptor (CAR)-T-cell immunotherapies to enhance their efficacy (Figure 5). Immunologically, intratumoral injection of mice with various soluble or encapsulated synthetic and natural CDNs (such as RR-CDG, the *R*,*R*-stereoisomer of the phosphorothioate analog c-di-GMP-S2 or RR-c-di-GMP-S2; RR-cyclic-di-guanine (CDG), *R*(P),R(P) dithio-c-di-GMP, ADU-S100, the *R*,*R*-stereoisomer of the phorphorothioate analog 2′3′-cAAMP-S2 or RR-2′3′-cAAMP or RR-CDA; c-di-GMP, and cGAMP-NPs) induced enhanced type I IFN signalling, potent local inflammatory responses (DCs, macrophages, NK cells, and CXCL10 and CCL5 responses), reprograming of macrophages from protumorigenic M2-like type toward M1-like type, and the activation of adaptive immunity (CD4 and CD8 T cells, and T-cell migration) to increase tumour immunogenicity [70,71,72,73,74,75,76]. As a result, the treatment reprogramed the tumor microenvironment toward a more immunogenic antitumor milieu and induced profound regression and necrosis of established tumours. In many cases, the treatment-induced substantial and durable systemic tumour-specific Th1 adaptive immunity is capable of rejecting distant metastases and providing long-lived immunologic memory [72,73,74,76]. The intratumoral CDN treatment also increased the response rate to T-cell receptor activation by checkpoint-modulating antibodies to CTLA-4, PD-1, 4-1BB, and OX40, resulting in significant increases in median survival time in animal models of melanoma (B16-F10 and YUMM1.7), breast adenocarcinoma (E0771), prostate cancer (TRAMP-C2), head and neck squamous cell cancers (SCCFVII) [72,73], HER-2^+^ breast tumours [76], glioblastoma multiforme [77], glioma [70], neuroblastoma [75], and 4T1 breast tumour [72,74,78,79,80,81]. In this regard, a single dose of cGAMP-liposomal nanoparticles (cGAMP-NP) was sufficient to modulate the tumour microenvironment for effective control of programmed death-ligand 1 (PD-L1)-insensitive basal-like triple-negative breast cancer and melanoma and, more significantly, prevented the formation of secondary tumours in mice [82]. Moreover, coadministration of c-di-GMP with implantable biopolymer devices to deliver CAR T cells directly to the surfaces of solid tumours improved the effectiveness of CAR T-cell therapy and protected against the emergence of escape variants in immunocompetent orthotopic mouse models of pancreatic cancer and melanoma [83].

Studies in mouse models of breast cancer have shown that intratumoral administration of ADU-S100 (an analog of c-di-AMP) alone was able to induce durable tumour clearance in 100% of parental FVB/N mice that are able to mount potent immune responses to the implanted neu-positive tumour cells, but the treatment only resulted in tumour clearance in 10% of the FVB/N-derived neu/N transgenic mice that have well-established peripheral immune tolerance to the endogenous neu antigen [76,81]. Coadministration of ADU-S100 with anti-PD-L1 and anti-OX40 antibodies significantly enhanced the clearance rate to 40% of neu/N mice and also induced abscopal immunity [76,81]. The efficacy of combination treatment was well correlated with globally enhanced ratios of CD8^+^ T cells to regulatory T cells, macrophages, and myeloid-derived suppressor cells, and downregulation of the M2 marker CD206 on tumour-associated macrophages in this model [81]. Unfortunately, ADU-S100 failed to demonstrate sufficient clinical benefit in clinical trials.

CDNs have also been used with other immunomodulators, such as CpG oligonucleotides (ODN) and monophosphoryl lipid A (MPLA), for cancer immunotherapy [85,86]. Systemic delivery of c-di-GMP/MPLA-encapsulated immuno-nanoparticles induced extensive upregulation of antigen-presenting cells (APCs) and NK cells in the blood and tumour tissue and resulted in significant therapeutic benefit in a mouse model of metastatic triple-negative breast cancer [85]. Compared with single treatments, intratumoral injection of both CpG ODN and cGAMP significantly reduced tumour size in mouse models of lymphoma and melanoma [86].

Although the precise events induced by CDNs in the tumour microenvironment remain to be determined, mechanistic studies using knockout and bone marrow chimeric mice suggest that both stromal and immune cells (bone marrow-derived TNFα) and their crosstalk are essential to achieve CDN-induced tumour regression and clearance [87]. In this regard, cGAMP-induced STING activation promoted the polarization of tumour-associated macrophages into proinflammatory subtypes in spontaneous gastric cancer in p53(+/−) mice and cell line-based xenografts and induced apoptosis of gastric cancer cells through IL-6R-Janus kinase (JAK)-IL-24 pathway [88]. The potential role of macrophages in the CDN-induced antitumor activities has also been analyzed in detail. Intratumoral injection of cGAMP induced the transient accumulation of macrophages in the tumor tissue in mouse models of 4T1 breast cancer, squamous cell carcinomas, CT26 colon cancer, and B16F10 melanoma. Clodronate liposome-mediated macrophage depletion impaired the antitumor effect of cGAMP treatment [89]. It has also been reported that c-di-AMP-induced NK cell-mediated tumor rejection involved type I IFN-mediated activation of NK cells and the enhanced expression of IL-15 and IL-15 receptors [90].

## 6. CDNs as Potent Vaccine Adjuvants for Systemic and Mucosal Immunization

The potential adjuvant function of CDNs was first demonstrated in mice with model antigens beta-galactosidase (beta-Gal) and ovalbumin (Ova). Subcutaneous immunization with c-di-GMP or c-di-AMP and beta-Gal or Ova elicited potent antigen-specific serum IgG titers, a balanced Th1/Th2 response, and strong in vivo cytotoxic T lymphocyte (CTL) responses (i.e., 30–60% of antigen-specific lysis) [91]. When immunized by mucosal routes (i.n. or sublingual), these vaccines also induced specific secretory IgA responses in multiple mucosal sites (such as lung and vagina) [49,92]. Similar results were also observed when bis-(3′,5′)-cyclic dimeric inosine monophosphate (c-di-IMP) or cGAMP was used [93,94].

Compared to other licensed and experimental adjuvants such as LPS, CpG ODN, and aluminum salt, c-di-GMP appears to be more potent in inducing both humoral and Th1 immune responses (higher IgG2a, IFN-γ, TNF-α, and CXCL10 responses) in mice and in non-human primate PBMC cultures [95]. Mice immunized with c-di-GMP also showed a more predominant germinal centre formation, indicative of long-term memory, than the mice immunized with LPS or CpG [95]. Similarly, recent studies have shown that s.c. immunization of mice with c-di-AMP and Ova resulted in significantly higher levels of Ova-specific IgG and stronger Th1 and IFNγ-producing CD8^+^ memory T-cell response than the mice immunized with poly(I:C)/CpG and Ova [96].

The potency and immune profiles induced by different CDNs and their derivatives have not been systemically compared in a head-to-head manner. Overall, it appears that all CDNs are capable of stimulating the host’s innate immune responses but often with subtle differences. Based on the antigen-specific IgG1 to IgG2a ratio and cytokine response profiles, earlier studies suggest that c-di-GMP induces a dominant Th1 response, cGAMP induces a balanced Th1/Th2 response, whereas c-di-AMP and c-di-IMP promotes a balanced Th1/Th2/Th17 response [49,92,93,94]. In murine BMDCs and human DCs, it has also been shown that cGAMP induced fewer IL-17 responses than c-di-AMP [94], but 3′3′-cGAMP stimulated stronger IFN gene activation than c-di-GMP [97]. Molecular dynamics simulation analysis showed that human STING binds more strongly to 3′3′-cGAMP than to c-di-GMP [98], although CDN derivatives capable of activating all human STING alleles and murine STING have been successfully synthesized [71]. In addition, Yan et al. have chemically synthesized several c-di-GMP analogs (mono- and bisphosphorothioate (c-di-GMP-S1 and -S2)) and the 2′-fluoro analog of c-di-GMP (2′-F-c-di-GMP) and compared their immunostimulatory properties in mice. Although the innate immune responses induced by these analogs were generally comparable to those induced by the parental c-di-GMP [99,100], the pulmonary inflammatory responses (neutrophils and chemokine CXCL1, CCL4, and CCL5) induced by the two phosphorothioate analogs (S1 and S2), particularly the c-di-GMP-S1, were much milder than the parent c-di-GMP [99]. On the other hand, 2′-F-c-di-GMP was effective when administered orally [100].

### 6.1. CDNs as Adjuvants for the Development of Vaccines against Infectious Diseases

The potential of CDNs as effective vaccine adjuvants for systemic and mucosal immunization against infectious diseases was first demonstrated in the development of vaccines against pneumococcal and *S. aureus* infections [48,62]. Intraperitoneal immunization of mice with c-di-GMP and pneumolysin toxoid (PdB) or pneumococcal surface protein A (PspA) induced significantly higher antigen-specific antibody responses and increased survival of mice after *S. pneumoniae* challenges, as compared to alum adjuvant [62]. Moreover, i.n. immunization of mice with pneumococcal surface adhesion A (PsaA) and c-di-GMP elicited strong antigen-specific serum IgG and secretory IgA responses at multiple mucosal surfaces and significantly reduced nasopharyngeal *S. pneumoniae* colonization in the immunized mice [61]. This study provided the first evidence of c-di-GMP as an effective mucosal adjuvant to induce protective immunity against mucosal bacterial infection. It appears that, in mice, c-di-GMP induces stronger immune responses and better protection against *S. pneumoniae* challenge than cGAMP-adjuvanted vaccine [65].

Many groups have subsequently explored the possibility of CDNs to improve the vaccine-induced immune profiles and the duration of immunity against a number of bacterial infections of human and veterinary importance (Table 1). Tuberculosis (TB), caused by *Mycobacterium tuberculosis*, remains a significant public health threat in many regions. *Bacillus Calmette-Guerin* (BCG), the live attenuated TB vaccine developed more than a century ago, is the most commonly used vaccine worldwide. However, BCG has varied protective efficiency in adults and safety concerns in the immunocompromised population. Several laboratories have attempted to use CDNs to enhance the current BCG vaccines or develop new, safe and effective subunit TB vaccines. Early efforts to introduce the c-di-GMP phosphodiesterase gene Rv1357c into BCG Pasteur failed to provide better protection against *M. tuberculosis* challenge than conventional BCG in mice [101]. Vaccination with another modified BCG strain (BCGΔBCG1419c) lacking the c-di-GMP phosphodiesterase gene BCG1419c induced a better activation of specific T lymphocytes, moderate protection, and improved resistance to infection reactivation in mice, as compared to vaccination with the conventional BCG vaccine [102]. Two groups have independently generated two recombinant BCG vaccines (rBCG-DisA), which overexpress the endogenous mycobacterial diadenylate cyclase (DisA) gene and produce high levels of c-di-AMP. Compared to conventional BCG vaccines, both rBCG-DisA vaccine candidates induced stronger immune responses, but one failed to provide additional protection against *M. tuberculosis* infection in mice [103], whereas the other showed significantly reduced lung weights, pathology scores, and *M. tuberculosis* burdens in guinea pigs [104].

When the CDN is used as an adjuvant for subunit TB vaccines, it enhances antigen-specific Th1 responses and elicits long-lasting protective immunity to *M. tuberculosis*. Immunization of mice with recombinant Rv3852 (H-NS) and cGAMP resulted in an enhanced antigen-specific IL-2 response by splenic lymphocytes, high serum levels of IL-2, IL-12p40, and TNF-α, and better protective efficacy (reduced bacterial counts in the spleen, but not in the lung) in mice challenged with *M. tuberculosis* H37Rv [133]. Similarly, s.c. immunization of a subunit TB vaccine consisting of a five *M. tuberculosis* antigen (antigen-85B, ESAT-6, Rv1733c, Rv2626c, and RpfD) fusion protein and a synthetic CDN (RR-CDG) induced protection equivalent to that induced by BCG vaccine. Interestingly, the protection in this study was STING-dependent but type I IFN-independent and correlated with an increased frequency of a subset of CXCR3-expressing T cells in the lung tissue. More significantly, i.n. immunization of this vaccine candidate induced protection superior to that induced by the BCG vaccine and elicited both Th1 and Th17 immune responses, the latter of which was correlated with enhanced protection [139]. On the other hand, i.n. immunization of mice with ESAT-6 and c-di-AMP failed to reduce the bacterial burdens after an i.v. challenge as compared to the mice immunized with ESTA-6 alone [119].

Failure to induce potent Th1 immune responses and long-term immunity and protect against nasal colonization and transmission of *Bordetella pertussis* by the current acellular pertussis (aP) vaccines have contributed to the recent global resurgence of pertussis (whooping cough). Intranasal immunization of mice with an experimental aP vaccine formulated with c-di-GMP and LP1569, a TLR2 agonist from *B. pertussis* (LP-GMP), induced potent antigen-specific Th1 and Th17 responses and conferred long-lasting protection (for 10 months) against nasal colonization and lung infection with *B. pertussis* [111]. The long-term protection against nasal colonization was correlated to IL-17-secreting respiratory tissue-resident memory CD4 T cells. Others have studied the adjuvant activity of cGAMP for parenteral and mucosal immunization with recombinant *Helicobacter pylori* urease A, urease B, and neutrophil-activating protein [132]. The gastric mucosal *H. pylori* colonization was significantly reduced in mice immunized by i.n. and, to a lesser degree, s.c. route, but not by the i.m. route. The protection efficacy appeared to be associated with the level of antigen-specific Th1 and particularly Th17 responses [132]. Moreover, Li et al. have demonstrated the potential of 2′-Fluoro-c-di-GMP as an oral vaccine adjuvant. Oral immunization of mice with *H. pylori* cell-free sonicate extract or flagellin proteins from *C. difficile* and *Listeria monocytogenes* in combination with 2′-Fluoro-c-di-GMP induced antigen-specific serum IgG and mucosal IgA responses and showed a significant reduction in the gastric bacterial colonization when mice were orally challenged with *H. pylori* [100].

The adjuvant function of CDNs has also been demonstrated in experimental vaccines against other bacterial and parasitic infections, including bacteria of antimicrobial resistance and biodefense significance. Subcutaneous immunization of mice with c-di-GMP and *S. aureus* antigen, clumping factor A (ClfA), or a nontoxic mutant staphylococcal enterotoxin C (mSEC) significantly reduced the bacterial burdens in the spleen and liver and improved the day 7 survival rates (87.5%) than those of the non-immunized mice (33.3%) after i.v., challenge with methicillin-resistant *S. aureus* [48,112]. Similarly, sublingual immunization of mice with *Bacillus anthracis* protective antigen (PA) and 3′3′-cGAMP induced comparable levels of PA-specific serum IgG and saliva IgA responses to those induced after immunization with PA and cholera toxin (CT), the golden standard mucosal adjuvant [138]. A single dose immunization with F1-V antigens of *Yersinia pestis* and RR-CDG induced 95% protection against a lethal challenge with *Y. pestis* CO92 within 14 days post-immunization [140]. Encapsulation of F1-V within nanoparticles further improved the efficacy and duration of the protection at a high challenge dose (≥7000 colony-forming units) in that 75% of immunized mice were protected from challenge at 182 days post-immunization [140]. Others have shown that the combination of the lipopeptide-based nano carrier system with c-di-AMP significantly reduced the antigen dose of a peptide vaccine and protected mice against a respiratory challenge with a lethal dose of a heterologous *S. pyogenes* strain [120]. Moreover, immunization of mice with several different recombinant antigens (a chimeric antigen Traspain, Tc52 or its N- and C-terminal domains) of *Trypanosoma cruzi* and c-di-AMP induced stronger protection against infection with higher survival rates, lower parasitemia, and reduced weight loss and chronic inflammation, compared to those immunized with other adjuvants [123,124,125]. Finally, c-di-GMP has also been explored for the development of live attenuated vaccines against salmonellosis (ΔXIII) [141,142] and mycoplasmal infections [126]. Under these circumstances, the CDNs were either overexpressed in the live attenuated vaccine or delivered together with cationic liposomes [126].

The promise of CDNs as systemic and mucosal adjuvants for influenza vaccines has been extensively studied with different influenza vaccines and antigens, including hemagglutinin (HA), neuraminidase (NA), matrix protein 2 ectodomain (M2e), and inactivated viruses. Overall, these studies have demonstrated CDNs as effective adjuvants by promoting potent, long-lasting mucosal and systemic humoral (IgA, IgG, IgG isotypes, and hemagglutination inhibition (HI) titers) and cellular immune responses (high frequencies of virus-specific polyfunctional CD4^+^ T cells producing one or more Th1 cytokines (IFN-γ, IL-2, TNF-α), and protection against challenges with homologous and heterologous viral strains (H1N1, H5N1, H3N2, H5N1, H7N9, and H9N2) in mice and ferrets [105,106,107,108,109,117,118,127].

Several studies have shown that the combination of CDNs with other adjuvants and vaccine delivery systems (such as chitosan, silica nanoparticle, acetylated dextran microparticles, and microneedle) further enhanced the immunogenicity, dose-sparing, and cross-clade protection of the experimental or licensed influenza vaccines [107,108,110,117,127,137]. In this regard, formulation of an H1N1 vaccine with pulmonary surfactant-biomimetic liposomes encapsulating cGAMP elicited cross-protection against distant H1N1 and heterosubtypic H3N2, H5N1, and H7N9 viruses for at least 6 months [130]. Similarly, immunization of mice with cGAMP-adjuvanted inactivated H7N9 vaccine also provided effective cross-protection against H1N1, H3N2, and H9N2 influenza viruses, induced significantly higher nucleoprotein-specific CD4^+^ and CD8^+^ T-cell responses in immunized mice and upregulated the IFN-γ and granzyme B expression in the lung tissue of mice in the early stages post a heterosubtypic virus challenge [127]. The adjuvant effect and safety of cGAMP were further confirmed in a pig model of intradermal immunization with the H5N1 influenza vaccine [129]. The extensive studies of CDNs in influenza vaccine development have also demonstrated the broad application of CDNs in multiple animal species (mice, rats, guinea pigs, ferrets, cattle, pigs, and chickens) by different immunization routes (i.n., i.m. intratracheal, intradermal, and sublingual).

However, whether CDNs can effectively enhance specific immune responses in aged or newborns remains to be determined [131,136]. Borriello et al. found that although cGAMP induced a comparable expression of surface maturation markers in newborn and adult BMDCs, cGAMP adjuvantation alone failed to increase HA-specific antibody responses in newborn mice [136]. However, immunization of puppies with cGAMP formulated with alum (Alhydrogel)-adjuvanted trivalent recombinant HA influenza vaccine (Flublok) significantly enhanced HA-specific IgG2a/c titers by six weeks of age [136]. Immunization with cGAMP+alum also enhanced IFNγ production by CD4^+^ T cells and increased both the absolute numbers and percentages of CD4^+^CXCR5^+^PD-1^+^ T follicular helper cells and GL-7^+^CD138^+^ germinal centre B cells. Similarly, immunization of an H1N1 influenza vaccine with cGAMP and saponin (Quil-A) significantly improved the survival (by 80–100%) of vaccinated 20-month-old mice, whereas cGAMP alone showed no significant improvement [143]. In this regard, recent studies found that targeting MoDCs with TNF fusion proteins restored the c-di-GMP adjuvantivity in aged mice [144].

The adjuvant activity of CDNs has also been demonstrated in vaccines against other viral infections, such as hepatitis B surface antigen (cGAMP) [135], HCV rE1E2 (c-di-AMP and archaeosomes) [121], HIV-1 reverse transcriptase (c-di-GMP) [113], mutant human papillomavirus (HPV) 16 E7 (E7GRG) protein (cGAMP and CpG-C ODN) [134], and foot and mouth disease vaccines in cattle and pigs [114]. In this regard, therapeutic immunization of female C57BL/6 mice with E7GRG formulated with cGAMP and CpG-C ODN induced significantly suppressed TC-1 tumour growth in mice [134].

### 6.2. CDNs as Adjuvants for the Development of Cancer Vaccines

The potential of CDNs in the development of vaccines against chronic, non-communicable diseases, particularly cancer, has also been investigated. It has been well recognized that effective therapeutic cancer vaccines require potent adjuvants to induce robust type I IFN and proinflammatory cytokine responses in the tumour microenvironment. CDNs, such as c-di-GMP, c-di-AMP, and cGAMP, are excellent adjuvants for the development of cancer vaccines because they have the potential to provide “two punches” to eliminate the tumour: activating immunogenic tumour cell death directly and inducing type 1 IFN production to enhance tumour-specific Th1 and CTL responses (Figure 5). Targeting both pathways is likely to result in a nearly complete local tumour regression and elimination of metastases [145]. Despite these rationales, relatively few studies have evaluated CDNs as a cancer vaccine adjuvant. This is perhaps due to the limited numbers of tumour-specific or associated antigens identified and available for cancer vaccine development. Chandra et al. reported that therapeutic immunization of metastatic breast cancer-bearing mice (4T1 model) with an attenuated *L. monocytogenes* (LM) expressing tumour-associated antigen MAGE-B (LM-Mb), followed by multiple low doses of c-di-GMP (0.2 μmol/L) treatment, resulted in almost complete elimination of all metastases. In this model, it appears that low doses of c-di-GMP significantly increased the production of IL-12 by myeloid-derived suppressor cells and improved antigen-specific T-cell responses, while high doses of c-di-GMP (range: 0.3–3 mmol/L) killed the 4T1 tumour cells directly by activating the tumour cell caspase-3 pathway [146].

CDNs have also been evaluated with a number of other cancer antigens and cancer vaccines, including MUC1 glycopeptide [147], a B16 melanoma peptide [148], and a mouse model of glioma [70] and EG7-Ova tumour [57]. In these applications, c-di-GMP and cGAMP functioned as potent immunostimulants and induced robust antitumor immunity and tumour regression. Shae et al. have recently developed a synthetic cancer nanovaccine platform (nanoSTING-vax) for efficient co-delivery of tumour peptide antigens and cGAMP to the cytosol of APCs [149]. Therapeutic immunization with this platform, in combination with immune checkpoint blockade, induced significant suppression of tumour growth, even complete tumour rejection, and long-lasting antitumor immune memory in multiple murine tumor models.

## 7. Novel Delivery Strategies to Enhance the Immunostimulatory and Adjuvant Function of CDNs

The in vivo efficacy of CDNs is limited by their physiochemical properties such as negative charges, hydrophilicity, short plasma half-life, and poor cellular targeting and membrane permeability. To overcome these challenges, researchers have developed several innovative strategies to transport CDNs more efficiently to the cell cytosol where STING is located (Table 2). These include in vivo synthesis of CDNs, nanoparticles, and microparticles and combination with other adjuvants and immunostimulators, etc. In this regard, the adjuvant activity of CDNs can be simply further enhanced or tailored by combination with other adjuvants and immunomodulators, particularly those that activate different immune signalling pathways, such as TLR agonists (CpG ODN, lipid A, and Pam_3_CSK), macrophage inducible C-type lectin (Mincle), saponin, and alum, etc. [97,114,150,151]. Others have attempted to synthesize c-di-GMP in vivo by transducing a diguanylate cyclase gene into the nonreplicating adenovirus serotype 5 (Ad5) vector, which produced enhanced amounts of c-di-GMP when expressed in mammalian cells in vivo [152,153]. Furthermore, cGAMP has been incorporated into viral particles such as lentivirus and herpesvirus virions [154]. The proof of concept of these approaches has been demonstrated in animal models with extracellular protein Ova, HIV-1 Gag peptides, or the *C. difficile* toxin B [152,153]. Quintana and colleagues have explored the feasibility to engineer a live *Lactococcus lactis* vector containing a single plasmid with both cdaA and tscf genes under different promoters, respectively, to simultaneously synthesize the adjuvant c-di-AMP as well as a heterologous vaccine antigen of interests in order to develop a simple and economical system for vaccine development [155]. However, control of the level and duration of the CDN expression by this approach can be a challenge since chronic STING activation has been involved and responsible for initiating certain inflammatory and autoimmune diseases due to type 1 IFN overproduction [4,156,157] In addition, sustained gene transcription is rigidly prevented by the host to avoid lethal STING-dependent proinflammatory disease by unknown mechanisms [158,159].

Substantial efforts have been made to develop novel delivery systems (such as different compositions of microparticles, liposomes, and nanoparticles, STINGel, and microneedles) to facilitate the efficient and targeted CDN delivery [162,168,169,170,171,172,173]. Compared with unformulated (free) CNDs, these formulations have a markedly prolonged half-life and local retention of CDNs within the tumour microenvironment, enhanced the local infiltration of CD11c^+^ DCs, F4/80^+^ macrophages, CD4^+^ T cells, and CD8^+^ T cells, stimulated APC activation (high expression of CD80, CD86 and MHC class I), induced the production of proinflammatory cytokines (type I and type II IFN, IL-1β, and IL-6), dramatically increased expansion and function of antigen-specific CD8^+^ T cells, and facilitated NK cell-mediated MHC-I non-restricted antitumor immunity [74,77,78,79,160,161,174].

## 8. Conclusions and Future Direction

Most of the recent research on the immunostimulatory and adjuvant function of CDNs has focused on the development of vaccines against bacterial and viral infections, particularly in influenza and tuberculosis. However, it is encouraging to note that CDNs have been increasingly used as vaccine adjuvants for some challenging infections and for their veterinary application. The identification of STING as a crucial pathway in cancer immunology generated a great interest in the research and development of CDNs for cancer immunotherapy and cancer vaccine adjuvants, which has led to several human clinical trials. As adjuvants for prophylactic vaccines for infectious diseases, CDNs induce potent humoral and cellular memory immune responses in the systemic and mucosal compartments. These immune responses often provide long-lasting immunity, cross-protection, and antigen dose-sparing, although more studies are needed to determine if CDNs are able to enhance the immunity in aged or newborns. As adjuvants for therapeutic cancer vaccines, CDNs induce potent antitumor immunity, including the activation of cytotoxic T cells, NK cells, and tumor-associated macrophages, and achieve durable regression in multiple mouse models of tumours.

Preclinical studies have shown that CDNs function in multiple animal species (mice, rats, guinea pigs, ferrets, chicken, pigs, and cattle). These studies also indicate that CDNs can be administered through different parenteral and mucosal routes to induce the desired systemic and local immune responses. Human cell culture and molecular modelling indicate that CDNs bind to human STING and function in vitro. CDNs have several inherent advantages as an ideal vaccine adjuvant, and these include the relative ease to synthesize and formulate, known mode of action, and a large amount of preclinical efficacy data. Several novel strategies and approaches, such as in vivo synthesis, chemical modifications, combination with other vaccine adjuvants and delivery systems, have been developed over the years to improve the cellular uptake of CDNs and facilitate their cytosolic delivery to further enhance their efficacy and application. In addition, several novel methods of enzymatic and microbiological synthesis of CDNs for scale-up have been developed. Among these, the delivery via in vivo synthesis is an attractive strategy, but appropriate control of their expression level and duration to prevent the development of autoimmune diseases will be a challenge.

Although an overwhelming number of studies have convincingly demonstrated the promise of CDNs as an effective and safe adjuvant for mucosal and systemic vaccination, a few studies have shown no or partial effect of CDNs. Intranasal immunization of mice with BtaF trimeric autotransporter of *B. suis* and c-di-AMP conferred significant protection against intragastric *B. suis* challenge but failed to protect against the respiratory challenge, despite the presence of vaccine-induced specific mucosal IgA and the production of high IFN-γ levels by lung cells [122]. Furthermore, immunization of pigs with c-di-AMP and naked self-amplifying replicon RNA (RepRNA) encoding influenza virus nucleoprotein and HA induced only low-level immune responses in 20% of pigs [175].

Despite the promising progress and more than 15 years of R&D efforts by the biomedical research communities and pharm/biotech industries, the research on CDNs as vaccine adjuvants seems stalled at the preclinical development stage. To date, there are no licensed CDN products for human clinical use. Several clinical trials of the leading CDNs for cancer immunotherapy have been conducted with disappointing results. The reasons for this remain unknown, but it was suggested that the heterogeneity of the human STING gene and the potential immunocompromised status and immunosenescence in clinical trial participants are possible contributing factors [176]. In this regard, single nucleotide polymorphism analysis showed the common presence of STING variants and mutations in the human population (ranged from 1.5% in R293Q to 20.4% in R71H-G230A-R293Q (HAQ) [177]). Therefore, further studies should be directed to dissect the differences in immune responses induced by different CDNs and their mode of action between human and experimental animal cells. In addition, more stringent preclinical research and development in clinically relevant animal models will be needed to promote the clinical application of these promising molecules. Finally, strategies to design “pan-STING agonists” as prophylactic vaccine adjuvants should be considered to ensure their function in a majority of the population.

## Figures and Tables

**Figure 1 vaccines-09-00917-f001:**
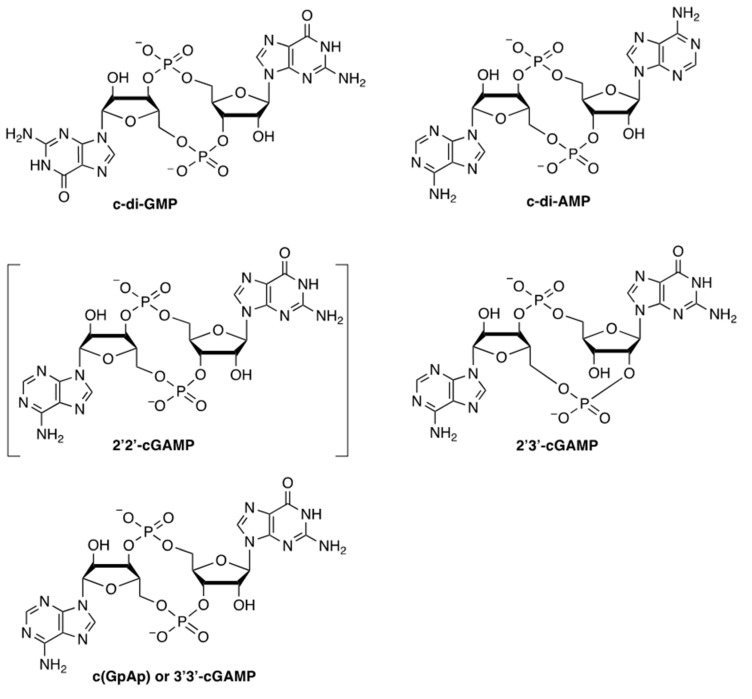
Chemical structures of naturally occurring cyclic dinucleotides known to date. The structure of 2′3′-cGAMP was originally misassigned as 2′2′-cGAMP shown in brackets.

**Figure 2 vaccines-09-00917-f002:**
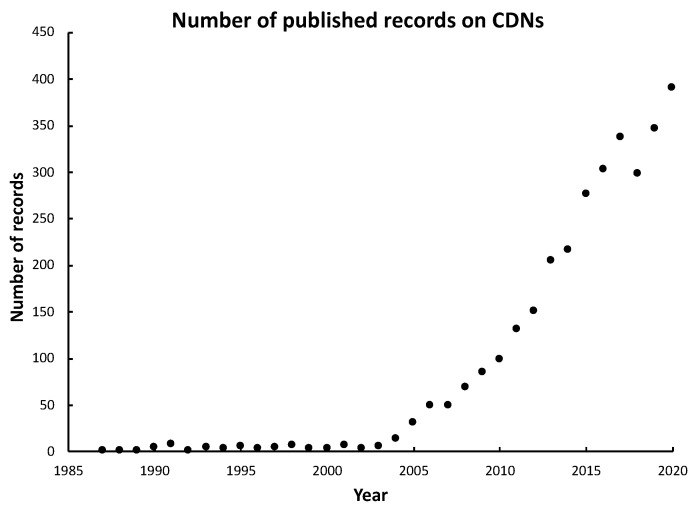
The number of published records on c-di-GMP and analogs retrieved (as of 20 May 2021) from the Web of Sciences with either topic or title containing one of the following terms: c-diGMP, c-di-GMP, cyclic di-GMP, cyclic diguanylic acid, c-GAMP, cGAMP, c-di-AMP, c-di-AMP, cyclic GMP-AMP, or cyclic dinucleotide, with duplicates removed.

**Figure 3 vaccines-09-00917-f003:**
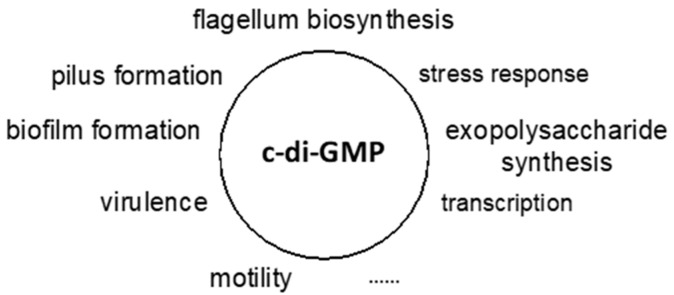
Processes regulated by c-di-GMP signalling networks.

**Figure 4 vaccines-09-00917-f004:**
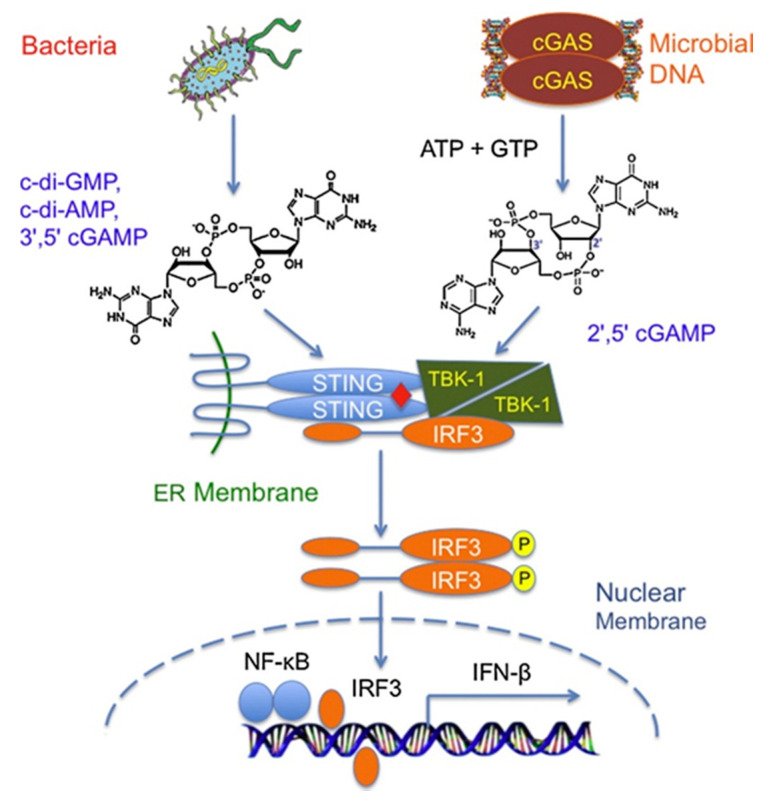
Innate immune sensing of bacterial cyclic dinucleotides and microbial DNA through the cGAS-STING pathway. Reproduced from Figure 1 of the work of [46] with permission from Copyright Clearance Center, Inc. ER: endoplasmic reticulum; TBK-1: TANK-binding kinase 1; IRF3: interferon regulatory factor 3; NF-κB: nuclear factor kappa B; IFN-β: interferon-β.

**Figure 5 vaccines-09-00917-f005:**
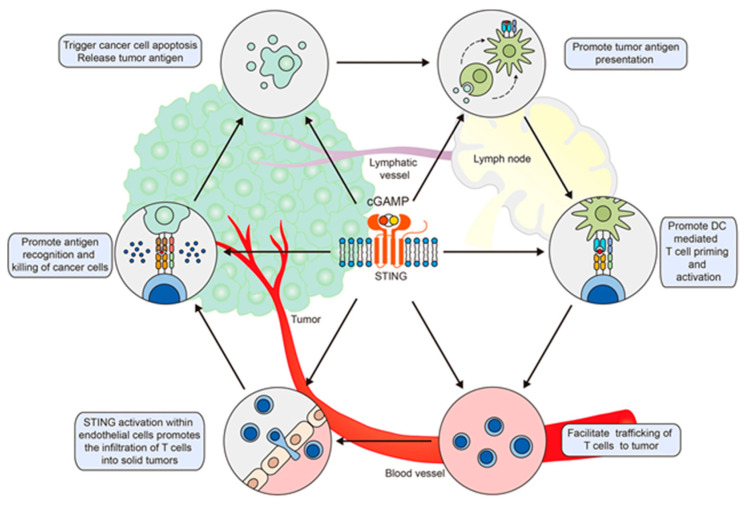
Activation of STING positively regulates each step of the cancer immunity cycle. Reproduced from Figure 2 of [84] with permission under the terms of the Creative Commons Attribution 4.0 International Licence (http://creativecommons.org/licenses/by/4.0/, accessed on 5 August 2021).

**Table 1 vaccines-09-00917-t001:** Selected application of cyclic dinucleotides (CDNs) as adjuvants in the development of vaccines against infectious diseases and cancers.

CDNs	Others	Species	Immunization Route	Antigens	Immunogenicity	Efficacy	References
c-di-GMP		Mice	SC	Beta-gal	Serum IgG, Th1/Th2		[91]
c-di-GMP		Mice	IN	Beta-gal, Ova	Serum IgG, mucosal IgA, Th1, CTL		[92]
c-di-GMP		Mice	IN, IM	Influenza H5N1 (A/Anhui/1/05)	Mucosal IgA, serum IgG, balanced Th1/Th2, high frequencies of multifunctional Th1 CD4^+^ cells. IN > IM		[105]
c-di-GMP		Mice	IN, sublingual, IM	Influenza H5N1 (A/Anhui/1/05)	Mucosal IgA, serum IgG, balanced Th1/Th2, high frequencies of splenic H5N1-specific multifunctional (IL-2^+^TNF-α^+^) CD4^+^ T cells. IN or SL > IM		[106]
c-di-GMP	Chitosan	Mice	IN	NIBRG-14 (H5N1) HA	Serum and local antibody responses, HI antibody, higher frequencies of virus-specific polyfunctional CD4^+^ T cells, more Th1 cytokines (IFN-γ, IL-2, TNF-α). 7.5 μg > 1.5 or 0.3		[107]
c-di-GMP	Silica nanoparticles (SiO_2_)	Mice	Intratracheal	Plant-produced H1N1 influenza HA	High systemic antibody responses, local IgG and IgA responses, local T-cell response (IL-2 and IFN-γ)		[108]
c-di-GMP		Mice, ferrets	IN	HA A/Indonesia/05/05 (H5N1)	IgG and IgA antibodies	Survived the viral challenge	[109]
c-di-GMP	microneedle	Mice	Microneedle skin patches	Influenza microneedle vaccine	Enhanced IgG, IgG subtypes, and cellular immune responses	100% survival rate and rapid weight recovery	[110]
c-di-GMP	*B. pertussis* LP1569 (a TLR2 agonist)	Mice	SC, IP, IN	Acellular pertussis vaccine	IFN-β, IL-12 and IL-23 responses, maturation of dendritic cells; IN: Th1 and Th17 responses; Th17 response and IL-17-secreting T_RM_ cells	IN: protection against nasal colonization and lung infection (sustained for at least 10 months)	[111]
c-di-GMP		Mice	SC	ClfA or mSEC (*S. aureus*)	High titers of IgG1, IgG2a, IgG2b and IgG3 compared to alum adjuvant mSEC > ClfA	Higher survival rates at day 7 with *S. aureus* challenge	[112]
c-di-GMP		Mice	IN, IP	*S. pneumoniae* PdB or PspA	IP induced higher antigen-specific antibodies	Significant decrease in bacterial load in lungs and blood; IP: increased survival compared to that with alum adjuvant	[62]
c-di-GMP		Mice	IN	*S. pneumonia* PsaA	Strong IgG and IgA	Significantly reduced nasopharyngeal colonization	[61]
c-di-GMP	DNA vaccine	Mice	ID	HIV-1 reverse transcriptase	Transient increase in IFN-γ production		[113]
c-di-GMP		Mice, cattle, pigs	IM	Foot and mouth disease vaccine	Early onset of high neutralizing antibody titers; long-lasting immune memory response		[114]
c-di-GMP		Mice	IN	Ghrelin-PspA	Specific antibody responses	Reduced body weight gain in diet-induced obesity mice	[115]
c-di-GMP		Rats, mice	IN	Angiotensin II type 1 receptor (AT1R) peptide conjugated with PspA	Specific antibody responses	Prevented the development of hypertension in spontaneously hypertensive rats; Sera protected mice against lethal pneumococcal infection	[116]
c-di-GMP and c-di-AMP		Mice	IN, sublingual	H5N1 virosomes	Local and systemic humoral and cellular immune responses; long-lasting immunity; dose-sparing	Effective protection against influenza H5N1	[117]
c-di-AMP		Mice	SC	Ova (soluble or DC targeted)	Serum IgG, Th1, CTL, and IFNγ-producing CD8 memory T-cell response, better than PolyI:C/CpG		[96]
c-di-AMP		Mice	IN	Influenza nucleoprotein	IgG and IgA responses, strong Th1 response (IFN-γ and IL-2)	Protection against A/Puerto Rico/8/34 (H1N1) virus	[118]
c-di-AMP		Mice	IN	ESAT-6	Innate and adaptive immune responses and regulated autophagy of macrophages	Protection against i.v. challenge similar to antigen alone group	[119]
c-di-AMP	Lipopeptide-based nano carrier systems	Mice	IN	T and B cell epitopes of *S. pyogenes* M protein	Antigen dose-sparing		[120]
c-di-AMP	Archaeosomes	Mice	IN-IN-IN, IM-IN-IN, or IM-IM-IM	HCV rE1E2	Induced more robust polyfunctional CD4^+^ T-cell responses		[121]
c-di-AMP		Mice	IN	BtaF trimeric autotransporter	Local and systemic antibody responses, central memory CD4^+^ T cells, and strong Th1 responses	Protection against intragastric, but not respiratory, challenge with *B. suis*	[122]
c-di-AMP	Conjugated at N-terminal of target antigen	Mice	IN	*T. cruzi* recombinant Tc52	Predominantly Th17 and Th1 immune responses	Better protection against infection with lower parasitemia and weight loss, and higher survival rates	[123]
c-di-AMP		Mice	IN	Traspain (trivalent *T. cruzi* antigen)	Primed Th1/Th17 immune response	Reduced parasite load and chronic inflammation	[124,125]
c-di-AMP	Poly(lactic-co-glycolic acid (PLGA) microparticles	Piglets	IM	Inactivated *M. hyopneumoniae* field isolate BA 2940-99	Strong innate immune responses and robust Th1 or Th17 responses		[126]
c-di-AMP or c-di-IMP		Mice	IN	Beta-gal, Ova	Serum IgG, mucosal IgA, Th1/Th2/Th17, CTL		[49,93]
cGAMP		Mice	IN, single dose	Inactivated whole virus H7N9 vaccine	Enhanced humoral, cellular, and mucosal immune responses. significantly higher nucleoprotein-specific CD4^+^ and CD8^+^ T-cell responses	Protection against a high lethal dose, effective cross-protection against H1N1, H3N2, and H9N2 influenza virus	[127]
cGAMP		Mice	IN	HA vaccine	Enhanced IgA, IgG, and T cell responses, including in nasal-associated lymphoid tissue		[128]
cGAMP		Pigs	ID	H5N1 and 2009 H1N1 pandemic influenza vaccines	Vigorous immune responses elicited with no overt skin irritation		[129]
cGAMP	Biomimetic liposomes	Mice, ferrets	IN	H1N1 vaccine	Vigorously augmented humoral and CD8^+^ T-cell responses, importance of alveolar epithelial cells in heterosubtypic immunity	Strong cross-protection against distant H1N1 and heterosubtypic H3N2, H5N1, and H7N9 viruses	[130]
cGAMP	Microneedle patches and saponin (Quil-A)	Aged mice	Microneedle patches	H1N1 vaccine	Increased IgG and IgG2a	Complete protection	[131]
cGAMP		Mice	IN, SC, IM	*H. pylori* urease A, urease B, and neutrophil-activating protein	Serum IgG and mucosal IgA, Th1, and particularly Th17 responses (IN, SC), IL-17 responses (IN)	Significantly reduced gastric mucosal *H. pylori* colonization (IN, SC)	[132]
cGAMP		Mice	IM	*M. tuberculosis* Rv3852 (H-NS)	High serum levels of IL-2, IL-12p40, and TNF-α, specific CD4 and CD8 T-cell responses	Reduced bacterial counts in the spleen but not in the lung after H37Rv challenge	[133]
cGAMP	CpG-C ODN	Mice	SC	HPV 16 E7 (E7GRG)	High IgG level and cell proliferation; IFN-γ and granzyme B levels	Suppressed TC-1 tumour growth	[134]
cGAMP		Mice	Not specified	HBsAg	Significantly enhanced humoral and cellular immune responses		[135]
cGAMP	Alum (Alhydrogel)	Mice (newborns)	IM	rHA influenza vaccine (Flublok)	Enhanced HA-specific IgG2a/c titers, IFNγ production by CD4^+^ T cells, increased T follicular helper cell and GL-7^+^CD138^+^ germinal centre B cell responses in newborns		[136]
3′3′ cGAMP	Acetalated dextran (Ace-DEX) microparticules	Mice	IM	Influenza matrix protein 2 (M2e)	Humoral and cellular responses, broadly protective immunity	Substantial cross-protection against distant H1N1 and heterosubtypic H3N2, H5N1, and H7N9 viruses	[137]
3′3′ cGAMP		Mice	SL	*Bacillus anthracis* protective antigen (PA)	Higher serum IgG and saliva IgA than PA + cholera toxin; Th1, Th2, and Th17 responses; rapid IFN-β and IL-10 responses in sublingual tissues and cervical lymph		[138]
RR-CDG		Mice	IN, SC	Antigen-85B, ESAT-6, Rv1733c, Rv2626c, and RpfD fusion protein	IN: significantly boosted BCG-induced immunity; elicited both Th1 and Th17 immune responses	IN: superior protection than BCG	[139]
RR-CDG	Nanoparticles	Mice	SC	*Y. pestis* F1-V	Induced rapid and long-lived protective immunity	100% protected from *Y. pestis* lethal challenge within 14 days post-immunization, and 75% protected at 182 days post-immunization	[140]
rBCG-disA-OE	Endogenous c-di-AMP	Guinea pigs	ID	BCG	Significantly stronger TNF-α, IL-6, IL-1β, IRF3, and IFN-β levels than BCG in murine macrophages culture	Significantly reduced lung weights, pathology scores, and bacterial counts in lungs vs. BCG	[104]
rBCG-DisA	Endogenous c-di-AMP	Mice	SC	BCG	Produced more IFN-γ, IL-2, and IL-10, stronger expression of H3K4me3 vs. BCG	No additional protection against *M. tuberculosis* infection	[103]
rBCG (BCGΔBCG1419c)		Mice	SC	BCG	Better activation of specific T-cell population vs. BCG	No difference in protection vs. BCG	[102]
rBCG-Rv1357c	c-di-GMP phosphodiesterase gene Rv1357c	Mice	ID	BCG	More phagocytosed vs. BCG	Similar protection against *M. tuberculosis* challenge vs. BCG	[101]

SC, Subcutaneous; Beta-gal, Beta-galactosidase; Th1/Th2, Type 1/type 2 T helper cells; IN, Intranasal; OVA, Ovalbumin; CTL, Cytotoxic T lymphocytes; IM, Intramuscular; IL, Interleukin; TNF-α, Tumor necrosis factor type; HI, Hemagglutination inhibition; IFN, Interferon; TLR, Toll-like receptors; IP, Intraperitoneal; Th17, T helper cells type 17; TRM cells, Tissue-resident memory T cells; ClfA, Clumping factor A; mSEC, mutant Staphylococcal enterotoxin C; PdB, genetic toxoid derivative of Pneumolysin; PspA, Pneumococcal surface protein A; PsaA, Pneumococcal surface adhesin A; ID, Intradermal; AT1R, Angiotensin II type 1 receptor; ESAT-6, 6 kDa Mycobacterium tuberculosis early secretory antigenic target; i.v, Intravenous; HCV, Hepatitis C virus; HPV, Human papillomaviruses; HBsAg, Hepatitis B surface antigen; BCG, Bacille Calmette-Guérin.

**Table 2 vaccines-09-00917-t002:** Selected examples of cyclic dinucleotides (CDNs) delivery systems.

CDNs	Deliver Systems	Function	Reference
CDN	Biodegradable, poly(beta-amino ester) nanoparticles	Cytosol delivery and 10-fold improved treatment potency in established B16 melanoma tumours with anti-PD-1 in mice compared to free CDN	[80]
c-di-GMP	Liposomes made of a synthetic, pH-sensitive lipid of high fusogenicity (YSK05)	Deliver into the cytosol and induced melanoma regression	[160,161]
c-di-GMP	Liposome nanoparticles	Targeting lymphatics and draining lymph nodes after parental injection	[162]
c-di-GMP and 3′3′-cGAMP	Cell-penetrating peptide	Enhanced both cellular delivery and biological activity of the c-di-GMP in murine splenocytes	[97]
c-di-GMP	Cytotoxic cationic silica nanoparticles	Local retention and potent melanoma growth inhibition	[79]
c-di-GMP	Mesoporous silica nanoparticle (immuno-MSN)	Brain delivery for treatment of glioblastoma multiforme	[77]
c-di-GMP	Rhodamine B isothiocyanate fluorescent mesoporous silica nanoparticles, synthesized and modified with poly(ethylene glycol) and an ammonium-based cationic molecule	“In situ vaccination” strategy; dramatic inhibition of 4T1 breast tumour growth in mice	[78]
c-di-GMP and c-di-AMP	Microneedles (MN)	Skin delivery better than SC and MN-based alum	[163]
c-di-AMP	Surfactant-based inverse micellar sugar glass nanoparticles (NPs)	Needle-free transfollicular antigen delivery from intact skin; better than polylactic-co-glycolic acid (PLGA) and chitosan-PLGA NPs	[164]
c-di-AMP	Cationic liposomes	Enhanced *M. hyopneumoniae*-specific antibody and T-cell responses	[126]
cGAMP	A polymersome delivery platform	Improved half-life of cGAMP, enhanced immune activation, and tumour growth suppression	[74]
cGAMP	Linear polyethyleneimine/hyaluronic acid hydrogels	Delivered into phagocytic macrophage cells	[165]
cGAMP	Cationic liposomes with varying surface polyethylene glycol levels	Cytosolic delivery and delivery to metastatic melanoma tumour sites and APCs	[166]
cGAMP	Microneedle skin patches (MNPs)	Increased IgG and IgG2a responses to influenza vaccines in aged mice (21-month-old), and full protection against challenge	[131]
cGAMP	Lipidoid nanoparticle (93-O17S-F)	Cytosolic delivery	[167]

PD-1, programmed cell death protein 1; SC, subcutaneous; APCs, antigen-presenting cells.

## Data Availability

Not applicable.

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
