# Peer review of "The Promise and Challenges of Cyclic Dinucleotides as Molecular Adjuvants for Vaccine Development"

_vaccines, 2021, doi:10.3390/vaccines9080917_

Round 1

Reviewer 1 Report

The review summarizes our current understanding of CDNs and their analogs as adjuvants for prophylactic vaccines against various infections or adjuvants for cancer immunotherapy.  It cites the literature extensively and offers overall progress in this field.  The review can be improved significantly should it address the following:

1). Effects of single nucleotide polymorphisms (SNPs) of STING on the adjuvant potentials of various agonists.  SNP analysis reveals the occurrence of R71H-G230A-R293Q (HAQ) mutations of the STING gene in 20.4% of the human population, R232H in 13.7%, G230A-R293Q (AQ) in 5.2%, and R293Q in 1.5%.  A pan-agonist of STING is needed for prophylactic vaccine adjuvant to ensure it works on a majority of people.

2). Figures like Fig. 5, 4, 3 are disconnected from the content of corresponding sections.  It is a loss of opportunity to use figures to offer potential connections among different dots made by different research groups, which is the major purpose of a review paper.  Copying a figure from literature hardly serves this purpose.

3). Authors should provide their own insight into the future direction of this field and the prospect of CDNs as adjuvants in the clinics.  

4). Typos throughout the review should be checked and corrected accordingly.

Reviewer 2 Report

  1. Yan and W. Chen. The promise and challenges of cyclic dinucleotides as molecular adjuvants for vaccine development

Cyclic dinucleotides (CDNs) are recognized as ubiquitous secondary signaling molecules in bacteria and eukaryotic cells with potent immunomodulatory functions. The discovery that CDNs can act as agonists of the innate immune response via signaling through the stimulator of interferon genes (STING) sensor has propelled their development as a promising new class of vaccine adjuvants. In the current manuscript, H. Yan and W. Chen present a comprehensive overview of the chemical structure, biosynthesis regulation, functions of CDNs and recent progress and future challenges for efficient delivery of CDNs as adjuvants for vaccines against infectious disease and cancer.

First, I would like to congratulate the authors on the excellent, well-written review. It was relatively easy to read and follow, especially considering I am not an expert in innate immunity or CDN biology. I was particularly fond of the historical perspective from discovering these secondary messengers to their prominence as potential adjuvants. Overall, I believe the manuscript should be published after minor modifications (mostly cosmetic and/or typographical errors).

Typographic errors

Line 44: ‘characteristics’ instead of ‘characters’.

Line 60: the text is missing the full description of acronym DisA. There are other acronyms missing a full description such as PilZ, PelD, LapD, VpsT, FleQ, Bcam1349, and so on.

Line 109: ‘as’ instead of ‘by’.

Line 221: “in human cells”. It is probably ok to remove as this section of the manuscript is related to in vivo stimulatory function of CDNs.

Line 237: ‘broiler’. Many readers may be unfamiliar with the term. Maybe the authors could replace it with chicken.

Lines 246-247: ‘encapsulated-cGAMP microparticles (MPs)’ instead of ‘microparticles (MPs) encapsulated cGAMP’.

Lines 249-252: This excerpt should probable moved to the in vitro stimulatory function of CDNs.

Lines 255-256: ‘immunotherapeutic’ instead of ‘immunotheraputic’.

Line 263: ‘responses’ instead of ‘resposnes’.

Line 283: ‘models’ instead of ‘modles’.

Lines 289-291 and 304-306: these sentences should be rewritten. It is difficult to gauge the difference between nontolerant parental FVB/N vs tolerant neu/N mice. Maybe authors want to expand on that for clarification. Instead of referring to EG-7 or B16 F10, it would be better to refer to the tumor types, such as ‘mouse models of melanoma and…’.

Line: 428: ‘adjuvant’ instead of ‘adjuvnat’.

Line 432: remove the ‘the’ in ‘when the mice were orally challenged…’.

Line 461: ‘influenza’ instead of ‘influenze’.

Line 503: DNA should probably be removed from ‘HIV-1 reverse transcriptase DNA’.

Lines 517-518: remove the ‘the’ in ‘elimination of the metastases…’.

Line 518: remove the ‘a in ‘relatively a few studies…’.

Line 609: ‘studies’ instead of ‘stuides’.

Lines 627-628: The sentence ‘Furthermore, the potential of CDNs as adjuvants for vaccines against autoimmune diseases and life style related diseases should be further explored’ makes no sense and should be removed.

Round 2

Reviewer 1 Report

no further comments although the authors did not address my concerns adequately about the description of each figure, which is dissociated with the description in the manuscript.